Journal of Data-centric Machine Learning Research (2025)        Submitted 12/20; Revised 04/14; Published 06/16

# The FIX Benchmark:
# Extracting Features Interpretable to eXperts

| | |
|---|---|
| **Helen Jin**⊕* | HELENJIN@SEAS.UPENN.EDU |
| **Shreya Havaldar**⊕* | SHREYAH@SEAS.UPENN.EDU |
| **Chaehyeon Kim**⊕* | CHAENYK@SEAS.UPENN.EDU |
| **Anton Xue**⊕* | ANTONXUE@SEAS.UPENN.EDU |
| **Weiqiu You**⊕* | WEIQIUY@SEAS.UPENN.EDU |
| **Helen Qu**★ | HELENQU@SAS.UPENN.EDU |
| **Marco Gatti**★ | MGATTI29@SAS.UPENN.EDU |
| **Daniel A. Hashimoto**† | DANIEL.HASHIMOTO@PENNMEDICINE.UPENN.EDU |
| **Bhuvnesh Jain**★ | BJAIN@PHYSICS.UPENN.EDU |
| **Amin Madani**‡ | AMIN.MADANI@UHN.CA |
| **Masao Sako**★ | MASAO@SAS.UPENN.EDU |
| **Lyle Ungar**⊕ | UNGAR@SEAS.UPENN.EDU |
| **Eric Wong**⊕ | EXWONG@SEAS.UPENN.EDU |

⊕*Department of Computer and Information Science, University of Pennsylvania, USA*

★*Department of Physics and Astronomy, University of Pennsylvania, USA*

†*Department of Surgery, Perelman School of Medicine, University of Pennsylvania, USA*

‡*Department of Surgery, University of Toronto, Canada*

**Reviewed on OpenReview:** *https://openreview.net/forum?id=BJnusBahD3*

**Editor:** Hugo Jair Escalante

## Abstract

Feature-based methods are commonly used to explain model predictions, but these methods often implicitly assume that interpretable features are readily available. However, this is often not the case for high-dimensional data, and it can be hard even for domain experts to mathematically specify which features are important. Can we instead automatically extract collections or groups of features that are aligned with expert knowledge? To address this gap, we present FIX (Features Interpretable to eXperts), a benchmark for measuring how well a collection of features aligns with expert knowledge. In collaboration with domain experts, we propose FIXScore, a unified expert alignment measure applicable to diverse real-world settings across cosmology, psychology, and medicine domains in vision, language, and time series data modalities. With FIXScore, we find that popular feature-based explanation methods have poor alignment with expert-specified knowledge, highlighting the need for new methods that can better identify features interpretable to experts.

**Keywords:** Interpretable Features, Explainability

---

* Equal contribution.

## 1 Introduction

Machine learning is increasingly used in domains like healthcare (Tjoa and Guan, 2019), law (Atkinson et al., 2020), governance (Meijer and Wessels, 2019), science (de la Torre-López et al., 2023), education (Holstein et al., 2018) and finance (Modarres et al., 2018). However, modern models are often black-box, which makes it hard for practitioners to understand their decision-making and safely use model outputs (Rai, 2019). For example, surgeons are concerned that blind trust in model predictions will lead to poorer patient outcomes (Hameed et al., 2023); in law, there are known instances of wrongful incarcerations due to over-reliance on faulty model predictions (Zeng et al., 2016; Wexler, 2017). Although such models have promising applications, their opaque nature is a liability in domains where transparency is crucial (Jacovi et al., 2021; Hong et al., 2020).

To address the pertinent need for transparency and explainability of their decision-making, the interpretability of machine learning models has emerged as a central focus of recent research (Arrieta et al., 2019; Saeed and Omlin, 2023; Räuker et al., 2023). A popular and well-studied class of interpretability methods is known as *feature attributions* (Ribeiro et al., 2016; Lundberg and Lee, 2017; Sundararajan et al., 2017). Given a model and an input, a feature attribution method assigns scores to input features that reflect their respective importance toward the model's prediction. A key limitation, however, is that the attribution scores are only as interpretable as the underlying features themselves (Zytek et al., 2022).

Feature-based explanation methods commonly assume that the given features are already interpretable to the user, but this typically only holds for low-dimensional data. With high-dimensional data like images and text documents, where the readily available features are individual pixels or tokens, feature attributions are often difficult to interpret (Nauta et al., 2023). The main problem is that features at the individual pixel or token level are often too granular and thus lack clear semantic meaning in relation to the entire input. Moreover, the important features are also domain-dependent, which means that different attributions are needed for different users. These factors limit the usefulness of popular feature attribution methods on high-dimensional data.

Instead of individual features, people tend to understand high dimensional data better in terms of semantic collections of low level features, such as regions in an image or phrases in a document. Moreover, for a feature to be useful, it should align with the intuition of *domain experts* in the field. To this end, an interpretable feature for high-dimensional data should have the following properties. First, they should encompass a grouping of related low-level features (e.g., pixels, tokens), thus creating high-level features that experts can more easily digest. Second, these low-level feature groupings should align with domain experts' knowledge of the relevant task, thus creating features with practical relevance. We refer to features that satisfy these criteria as **expert features.**

But how can we obtain such features? In practice, this process is left to domain experts to identify and provide such features for individual tasks. Although experts often have a sense of what the expert features should be, formalizing such features is often non-trivial and difficult. Moreover, besides formalizing, manually annotating expert features can also be expensive and labor-intensive. Towards obtaining high-quality features, we ask the following question:

Can we automatically measure how well features align with expert knowledge?

| | Implicit Expert Features | | | | Explicit Expert Features | |
|---|---|---|---|---|---|---|
| | Cosmology | | Psychology | | Medicine | |
| **Dataset** | Mass Maps | Supernova | Multilingual Politeness | Emotion | Chest X-Ray | Cholecystectomy |
| **Input (x)** | mass map image | simulated astronomical time-series data | conversation snippet | Reddit comment | chest X-ray image | video surgery image |
| **Output (y)** | energy density $\Omega_m$, matter fluctuation $\sigma_8$ | astronomical sources (e.g. supernova) | politeness level | emotion | pathology | safe/unsafe zone |
| **# Examples** | 110,000 | 7,848 | 22,800 | 58,000 | 28,868 | 1,015 |
| **Expert Features** | voids, clusters | linear consistent wavelengths | lexical categories | Russell's circumplex model | anatomical structures | organ structures |
| **Input Example** | | | I was running my spellchecker and totally didn't realize that this was a vandalized page. Please accept my apology. I will spellcheck a little slower next time. | This was potentially the most dangerous stunt I have ever seen someone do. One minor mistake and you die. | | |
| **Examples of Expert Features** | | | | | | |
| **Adapted From** | [Kacprzak et al., 2023] | [Team et al., 2018] | [Havaldar et al., 2023a] | [Demszky et al., 2020] | [Majkowska et al., 2020] | [Madani et al., 2022] |

Figure 1: The FIX benchmark contains 6 datasets across a diverse set of application areas, data modalities, and dataset sizes. For each dataset, we show an example of an input and some example expert features for that input.

To this end, we present the FIX benchmark, a unified evaluation measuring feature interpretability that can capture each individual domain's expert knowledge. We propose a class of metrics called the FIXScore and a collection of real-world datasets with expert-designed features.

Our goal is to guide the development of new methods to produce interpretable features by introducing a unified evaluation metric for the expert interpretability of feature groups. The FIX datasets (summarized in Figure 1) collectively encompass a diverse array of real-world settings (cosmology, psychology, and medicine) and data modalities (vision, language, and time-series): abdomen surgery safety identification (Madani et al., 2022), chest X-ray classification (Lian et al., 2021), mass maps regression (Kacprzak et al., 2023), supernova classification (Željko Ivezić et al., 2019), multilingual politeness classification (Havaldar et al., 2023a), and emotion classification (Demszky et al., 2020; Havaldar et al., 2023b). The challenge lies in unifying all 6 different real-world settings and 3 different data modalities into a *single* framework. We achieve this with our proposed expert alignment measure FIXScore, allowing for a benchmark that does not overfit any particular domain. To our knowledge, while previous work has identified the need for interpretable features (Zytek et al., 2022; Doshi-Velez and Kim, 2017), a benchmark that measures the interpretability of features for real-world experts does not yet exist. The FIX benchmark accomplishes this and also serves as a basis for studying, constructing, and extracting expert features. In summary:

1. In collaboration with domain experts, we develop the **FIX** benchmark, a collection of 6 curated datasets with metrics for evaluating the explanation inheritability of high-level

features. Our datasets are taken from real-world settings and covers diverse modalities spanning images, text, and time-series data. [*]

2. We introduce a general feature evaluation metric, FIXSCORE, that unifies the different real-world settings of cosmology, psychology, and medicine into a single framework. The criteria for what made features interpretable in each domain were closely informed by real domain experts.

3. We evaluate commonly used techniques for extracting higher-level features and find that existing methods score poorly on FIXSCORE, highlighting the need for developing new general-purpose methods designed to automatically extract expert features.

## 2 Related Work

**Interpretability.** Interpretability in machine learning is a multifaceted concept that encompasses algorithmic transparency (Shin and Park, 2019; Rader et al., 2018; Grimmelikhuijsen, 2023), explanation methods (Marcinkevičs and Vogt, 2023; Havaldar et al., 2023c), and visualization techniques (Choo and Liu, 2018; Spinner et al., 2019; Wang et al., 2023), among other aspects. In this work, we focus on feature-level interpretability, a central topic in interpretability research (Hong et al., 2020; Nauta et al., 2023). Feature-based methods are popular because they are believed to offer simple, adaptable, and intuitive settings in which to analyze and develop interpretable machine learning workflows (Molnar et al., 2020). We refer to (Nauta et al., 2023; Dwivedi et al., 2023; Weber et al., 2023) and the references therein for extensive reviews on feature-based explanations.

**Application-grounded Evaluation.** Chaleshtori et al. (2024) extend the work of Doshi-Velez and Kim (2017) to propose a comprehensive taxonomy of evaluating explanations. Notably, this includes *application-grounded evaluations*, which broadly seek to measure the efficacy of feature-based methods in settings with human users and realistic tasks, such as AI-assisted decision-making. However, the available literature on application-grounded evaluations is sparse: Chaleshtori et al. (2024) reviewed over 50 existing NLP datasets and found that only four were suitable for application-grounded evaluations (DeYoung et al., 2019; Wadden et al., 2020; Koreeda and Manning, 2021; Malik et al., 2021). A principal objective of the FIX benchmark is to provide an application-grounded evaluation of feature-based explanations in real-world settings.

**Feature Generation.** Because high-quality and interpretable features may not always be available, there is interest in automatically generating them by combining low-level features (Nargesian et al., 2017; Erickson et al., 2020; Zhang et al., 2023a). Notably, Zhang et al. (2023a) propose a method for tabular data using the expand-and-reduce framework (Kanter and Veeramachaneni, 2015). However, existing generation methods do not necessarily produce interpretable features, and most works focus on tabular data. The FIX benchmark aims to address these limitations by providing a setting in which to study and develop methods for interpretable feature generation across diverse problem domains.

**XAI Benchmarks.** There exists a suite of benchmarks for explanations that cover the properties of faithfulness (or fidelity) (Zhou et al., 2021; Agarwal et al., 2022), robust-

---

[*] Packaged libraries of code, hugging face data loaders and updates are available at `https://brachiolab.github.io/fix/`

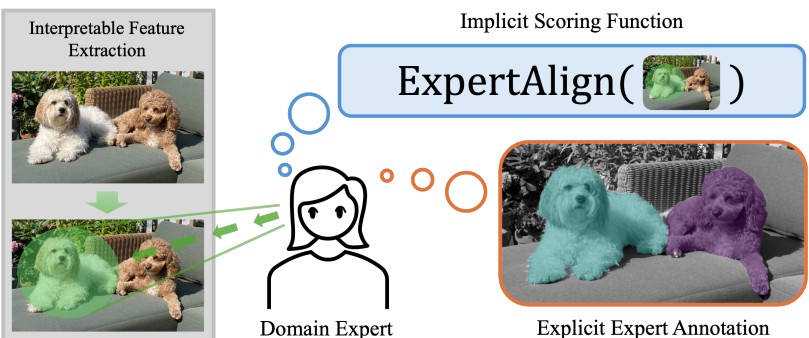

Figure 2: The FIX benchmark allows measuring alignment of extracted features with expert features in different domains, either implicitly with a scoring function or explicitly with expert annotations.

ness (Alvarez-Melis and Jaakkola, 2018; Agarwal et al., 2022), simulatability (Mills et al., 2023), fairness (Fel et al., 2021; Agarwal et al., 2022), among others. Quantus (Hedström et al., 2023), XAI-Bench (Liu et al., 2021), OpenXAI (Agarwal et al., 2022), GraphXAI (Agarwal et al., 2023), and ROAR (Hooker et al., 2019) are notable open-source implementations that evaluate for such properties. CLEVR-XAI (Arras et al., 2022) and Zhang et al. (2023b) provide benchmarks that combine vision and text. ERASER (DeYoung et al., 2019) is a popular NLP benchmark that unifies diverse NLP datasets of human rationales and decisions. In general, however, there is a lack of interpretability benchmarks that evaluate feature interpretability in real-world settings — a gap we aim to address with the FIX benchmark.

## 3 Expert Feature Extraction

Feature-based explanation methods require interpretable features to be effective. For example, surgeons communicate safety in surgery with respect to key anatomical structures and organs, which are interpretable features for surgeons (Strasberg and Brunt, 2010; Hashimoto et al., 2019). These interpretable features are a key bridge that can help surgical AI assistants communicate effectively with surgeons. However, ground-truth annotations for such interpretable features are often expensive and hard to obtain, as they typically require trained experts to manually annotate large amounts of data. This bottleneck is not unique to surgery, and such challenges motivate us to study the problem of extracting *features interpretable to experts*, or what we call expert features.

Consider a task with inputs from $\mathcal{X} \subseteq \mathbb{R}^d$ and outputs in $\mathcal{Y}$. In the example of surgery, $\mathcal{X}$ may be the set of surgery images, and $\mathcal{Y}$ is the target of where it is safe or unsafe to operate. We model a higher-level expert feature of input $x \in \mathcal{X}$ as a subset of features represented with a binary mask $g \in \{0,1\}^d$, where $g_i = 1$ if the $i$th feature is included and $g_i = 0$ otherwise. In surgery, for example, a good high-level feature is one that accurately selects a key anatomical structure or organ from an input $x$. The objective of interpretable feature extraction is to find a set of masks $\hat{G} \subseteq \{0,1\}^d$ that effectively approximates the

expert features of $x$. That is, each binary mask $\hat{g} \in \hat{G}$ aims to identify some subset of features meaningful to experts.

However, given a candidate subset of features, how can we judge whether the resulting subset is actually meaningful to experts? To analyze and evaluate potential expert features, we adopt the following **key guiding principle**: expert features should be ***designed by experts, for experts***. Specifically, to ensure broad utility to experts in real world problems, we have designed the FIX benchmark to satisfy the following three properties:

1. **Formulated by Experts**: Desirable expert features and their corresponding evaluation metrics should be developed by experts and be widely-accepted in their field. In all settings, we work directly with experts to ensure that all of the FIX datasets and their expert features are well supported and accepted in each domain.

2. **Misalignment of Models and Experts**: We focus the FIX benchmark on settings where experts by default reason with respect to expert features, but machine learning models typically use low level features. This mismatch is a major communication barrier when explaining model predictions to experts. The FIX settings span problems in medicine, scientific discovery, and social science where experts regularly communicate via expert features, such as organs in surgery, but models are trained in high dimensional inputs, such as high resolution images.

3. **Measure Algorithmic Progress in Expert Feature Extraction:** The ultimate goal of this benchmark is to guide the develop of novel expert feature extraction methods. to ensure that algorithms are of use to the broader scientific community, solutions should not be overly tailored to any single task. The FIX settings are designed to span a variety of machine learning modalities (vision, language, and time series) and learning problems (clarification, regression, and segmentation).

In contrast, existing interpretability benchmarks do not closely tie the features to expert knowledge. For example, CLEVR-XAI (Arras et al., 2022), ERASER (DeYoung et al., 2019), and ToolQA (Zhuang et al., 2023) benchmarks are built synthetically or are typical machine learning benchmarks that do not necessarily align with expert knowledge in practical domains. Other benchmarks, such as Ismail et al. (2020), DRAC (Qian et al., 2024), and FIND (Schwettmann et al., 2024) are task-specific and do not measure general algorithmic progress across domains.

### 3.1 Measuring Alignment of Extracted Features with Expert Features

Suppose we are given a function $\textsc{ExpertAlign}(\hat{g}, x) \in [0, 1]$ that measures how expert-interpretable a group $\hat{g} \in \{0, 1\}^d$ is for input $x \in \mathbb{R}^d$. Such alignment functions for individual groups are common in related tasks, such as in word semantics (Mathew et al., 2020), segmentation (Cordts et al., 2016; Abu Alhaija et al., 2018) or object detection (Everingham et al., 2010; Lin et al., 2014) etc. The challenge in designing FIXSCORE is to extend $\textsc{ExpertAlign}$ to a *set* of groups $\hat{G} \subseteq \{0, 1\}^d$ while ensuring that individual low-level features are well-covered by $\hat{G}$. To do this, we first define how well each low-level feature

$i = 1, \ldots, d$ aligns with respect to $\hat{G}$ and $x$ as follows:

$$\text{FEATUREALIGN}(i, \hat{G}, x) = \begin{cases} 0, & \text{if } \hat{G}[i] = \emptyset \\ \dfrac{1}{|\hat{G}[i]|} \displaystyle\sum_{\hat{g} \in \hat{G}[i]} \text{EXPERTALIGN}(\hat{g}, x), & \text{otherwise} \end{cases} \quad (1)$$

where $\hat{G}[i] = \{\hat{g} \in \hat{G} : i \in \hat{g}\}$ are the groups of $\hat{G}$ that cover feature $i$. This measures how well, on average, each covering group of $i$ aligns with the expert criteria of interpretability. This is to promote that each group of $\hat{G}[i]$ usefully contributes towards the alignment metric. We then extend FEATUREALIGN to all the low-level features to define:

$$\text{FIXSCORE}(\hat{G}, x) = \frac{1}{d} \sum_{i=1}^{d} \text{FEATUREALIGN}(i, \hat{G}, x) \quad (2)$$

where we note that FIXSCORE is parametrized by the particular choice of EXPERTALIGN function. FIXSCORE can thus be thought of as an average of averages: the expert alignment for each individual feature $i = 1, \ldots, d$ is averaged over all covers $\hat{G}[i]$. As a result, this metric has two key strengths regarding feature coverage:

1. **Duplication Invariance at Optimality.** If one extracts perfect expert features (i.e., FIXSCORE$(\hat{G}, x) = 1$ for some $\hat{G}$ and $x$), the FIXSCORE cannot be increased further by duplicating expert features. This property ensures that the score cannot be trivially inflated with repeated features.

2. **Encourages Diversity of Expert Features.** Since the score aggregates a value for each feature from $i = 1, \ldots, d$, adding a new expert feature that does not yet overlap with already extracted features is always beneficial.

The use of a generic expert alignment function enables the FIXSCORE to accommodate a diverse set of applications which fulfills the first desiderata of domain agnostic. To satisfy the third desideratum of expert alignment, FIXSCORE includes an expert alignment function customized by experts for each domain. There are two main ways one can specify the EXPERTALIGN function: *implicitly* with a score specified by an expert or *explicitly* with annotations from an expert, as shown in Figure 2.

**Case 1: Implicit Expert Alignment.** Suppose we do not have explicit annotations of expert features for ground truth groups. In this case, we use implicit expert features defined indirectly via a scoring function that measures the quality of an extracted feature. The exact formula of the score is specified by an expert and will depend on the domain and task. Implicit expert features have the advantage of potentially being more scalable than features manually annotated by experts.

**Case 2: Explicit Expert Alignment.** In the case where we do have annotations for expert features $G^\star$, we can use a standardized expression for the FIXSCORE that measures the best possible intersection with the annotated expert features. Then, the expert alignment score of a feature group $\hat{g}$ is

$$\text{EXPERTALIGN}(\hat{g}, x) = \max_{g^\star \in G^\star(x)} \frac{|\hat{g} \cap g^\star|}{|\hat{g} \cup g^\star|} \quad (3)$$

and $|\cdot|$ counts the number of ones-entries, and $\cap$ and $\cup$ are the element-wise conjunction and disjunction of two binary vectors, respectively. In other words, in the explicit case where the ground-truth expert features are known, alignment amounts to finding the best IoU score among all the expert-defined features $G^\star$. Matching intuition, FIXSCORE attains its optimal value at $\hat{G} = G^\star$:

**Theorem 1.** *In the explicit case where $G^\star$ is known and has full coverage (for all features $i = 1, \ldots, d$, there exists $g^\star \in G^\star$ such that $i \in g^\star$), we have $FIXSCORE(G^\star, x) = 1$ for all $x$.*

In this benchmark, the Mass Maps, Supernova, Multilingual Politeness, and Emotion datasets are examples of the implicit features case. On the other hand, the Cholecystectomy and Chest X-ray datasets are examples of the explicit expert features case.

Our goal in FIX is to benchmark general-purpose feature extraction techniques that are *domain agnostic* and do not use the FIXSCORE during training. Instead, benchmark challengers can use neural network models trained on the end-to-end tasks to automatically extract features without explicit supervision, which we release as part of the benchmark and discuss further in Appendix B. Annotations for expert features are too expensive to collect at scale for training, while implicit features are by no means comprehensive. The FIX benchmark is intended for evaluation purposes to spur research in general purpose and automated expert feature extraction.

## 4 FIX Datasets

To develop the FIX benchmark, we curated datasets for expert features designed by experts in accordance with the properties discussed in Section 3. In this section, we briefly describe each FIX dataset in Figure 1. For each dataset, we provide an overview of the domain task and the problem setup. We then introduce the key expert alignment function that measures the quality of an expert feature, and explain why certain properties incorporated in the expert alignment function are desirable to experts.

### 4.1 Mass Maps Dataset

**Motivation.** A major focus of cosmology is the initial state of the universe, which can be characterized by various cosmological parameters such as $\Omega_m$, which relates to energy density, and $\sigma_8$, which pertains to matter fluctuations (Abbott et al., 2022). These parameters influence what is observable by mass maps, also known as weak lensing maps, which capture the spatial distribution of matter density in the universe. Although mass maps can be obtained through the precise measurement of galaxies (Jeffrey et al., 2021; Gatti et al., 2021), it is not known how to directly measure $\Omega_m$ and $\sigma_8$. This has inspired machine learning efforts to predict the two cosmological parameters from simulations (Ribli et al., 2019; Matilla et al., 2020; Fluri et al., 2022). However, it is hard for cosmologists to gain insights into how to predict $\Omega_m$ and $\sigma_8$ from black-box ML models.

**Problem Setup.** Our dataset contains clean simulations from CosmoGridV1 (Kacprzak et al., 2023). Each input is a one-channel image of size $(66, 66)$, where the task is to predict $\Omega_m$ and $\sigma_8$. Here, $\Omega_m$ is the average energy density of all matter relative to the total energy density, including radiation and dark energy, while $\sigma_8$ describes fluctuations in the

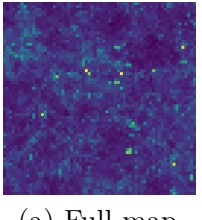 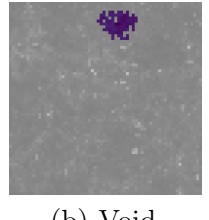 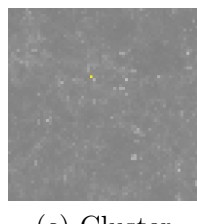

(a) Full map            (b) Void            (c) Cluster

Figure 3: An example with expert features for Mass Maps Regression, showing (a) the full map, (b) a feature with 100% void, and (c) a feature with 100% cluster. Voids are under-dense large regions that appear to be dark, and clusters are over-dense regions that appear as bright dots. The purity scores for both void and cluster are 1. We gray-out the pixels not selected in each feature.

distribution of matter (Abbott et al., 2022). The dataset has contains train/validation/test splits of sizes 90,000/10,000/10,000, respectively.

**Expert Features.** When inferring $\Omega_m$ and $\sigma_8$ from the mass maps, we aim to discover which cosmological structures most influence these parameters. Two types of cosmological structures in mass maps known to cosmologists are voids and clusters (Matilla et al., 2020). An example is illustrated in Figure 3, where voids are large regions that are under-dense relative to the mean density and appear as dark, while clusters are over-dense and appear as bright dots.

To quantify the interpretability of an expert feature in the mass maps, we develop an implicit expert alignment scoring function. Intuitively, a group that is purely void or purely cluster is more interpretable in cosmology, while a group that is a mixture is less interpretable. We thus develop the purity metric based on the entropy among void/cluster pixels (Zhang et al., 2003) weighted by the ratio of interpretable pixels in the expert feature. We give additional details in Appendix A.1.

$$\text{EXPERTALIGN}(\hat{g}, x) = \text{Purity}_{vc}(\hat{g}, x) \cdot \text{Ratio}_{vc}(\hat{g}, x) \tag{4}$$

### 4.2 Supernova Dataset

**Motivation.** The astronomical time-series classification, as mentioned in (Team et al., 2018), involves categorizing astronomical sources that change over time. Astronomical sources include transient phenomena (e.g., supernovae, kilonovae) and variable objects (e.g., active galactic nuclei, Mira variables). This task analyzes simulation datasets that emulate future telescope observations from the Legacy Survey of Space and Time (LSST) (Željko Ivezić et al., 2019). Given the vastness of the universe, it is essential to identify the time periods that have the most significant impact on the classification of astronomical sources to optimize telescope observations. Time periods with no observed data are less useful. To avoid costly searching over all timestamps for high-influence time periods, we aim to identify significant timestamps that are linearly consistent in specific wavelengths.

**Problem Setup.** We take parts of the dataset from the original PLAsTiCC challenge (Team et al., 2018). The input data are simulated LSST observations comprising four columns: observation times (modified Julian days), wavelength (filter), flux values, and flux error. The

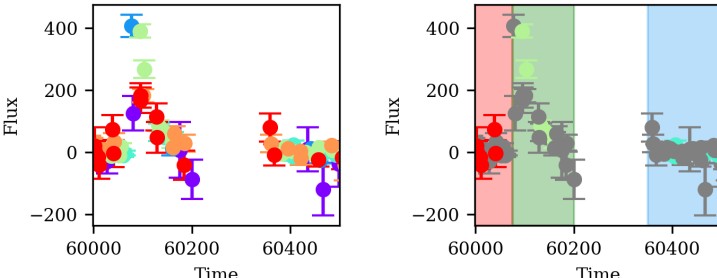

Figure 4: An example with expert features for supernova classification, showing (left) the original time-series dataset and (right) an example of the interpretable expert feature group. We highlight the expert feature groups with the highest ExpertAlign scores.

dataset encompasses 7 distinct wavelengths that work as filters, and the flux values and errors are recorded at specific time intervals for each wavelength. The classification task is to predict whether or not each of 14 different astronomical objects exists. The supernova dataset contains train/validation/test splits of sizes 6274/728/792, respectively.

**Expert Features.** A feature with linearly consistent flux for each wavelength is considered more interpretable in astrophysics. An illustration of expert features used for supernova classification is presented in Figure 4. This example showcases the flux value and error for various wavelengths, each represented by a different color. We colored the timestamp of expert features with the wavelength color with the highest linear consistency score. For timestamps where there are no data points, we do not recognize them as expert features. We create a linear consistency metric to assess the expert alignment score of a proposed feature in the context of a supernova. Our linear consistency metric uses $p$, the percentage of data points that display linear consistency, penalized by $d$, the percentage of time stamps containing data points:

$$\text{ExpertAlign}(\hat{g}, x) = \max_{w \in W} p(\hat{g}, x_w) \cdot d(\hat{g}, x_w). \tag{5}$$

where $W$ is the set of unique wavelength. Further details are provided in Appendix A.2.

### 4.3 Multilingual Politeness Dataset

**Motivation.** Different cultures express politeness differently (Leech, 2007; Pishghadam and Navari, 2012). For instance, politeness in Japan often involves acknowledging the place of others (Spencer-Oatey and Kádár, 2016), whereas politeness in Spanish-speaking countries focuses on establishing mutual respect (Placencia and Garcia-Fernandez, 2017). Therefore, grounding interpretable features that indicate politeness is *language-dependent*. Previous work from Danescu-Niculescu-Mizil et al. (2013) and Li et al. (2020) use past politeness research to create lexica that indicate politeness/rudeness in English and Chinese, respectively. A lexicon is a set of categories where each category contains a curated list of words. For instance, the English politeness lexicon contains categories like *Gratitude*: "appreciate", "thank you", et cetera, and *Apologizing*: "sorry", "apologies", etc. Havaldar et al. (2023a) expand on these theory-grounded lexica to include Spanish and Japanese.

| Example | Expert Features with High Alignment |
|---|---|
| *[Politeness]* I was running my spellchecker and totally didn't realize that this was a vandalized page. Please accept my apology. I will spellcheck a little slower next time. | $g_1 =$ `I, my, I`
$g_2 =$ `spellchecker, vandalized, little, slower`
$g_3 =$ `will`
$g_4 =$ `my, apology` |
| *[Emotion]* This was potentially the most dangerous stunt I have ever seen someone do. One minor mistake and you die. | $g_1 =$ `dangerous, die`
$g_2 =$ `potentially, minor`
$g_3 =$ `mistake, stunt`
$g_4 =$ `I, someone, you` |

Table 1: Examples and expert features with high expert alignment for Multilingual Politeness (top) and Emotion (bottom) settings. These expert features correspond to low distance within the emotion circumplex and high similarity with politeness lexica, respectively.

**Problem Setup.** The multilingual politeness dataset from (Havaldar et al., 2023a) contains 22,800 conversation snippets from Wikipedia's editor talk pages. The dataset spans English, Spanish, Chinese, and Japanese, and native speakers of these languages have annotated each conversation snippet for politeness level, ranging from -2 (very rude) to 0 (neutral) to 2 (very polite).

**Expert Features.** When extracting interpretable features for a task like politeness classification across multiple languages, it is useful to ground these features using prior research from communication and psychology. If extracted politeness features from an LLM are interpretable and domain-aligned, they should match what psychologists have determined to be key politeness indicators. Examples of expert-aligned features are shown in Table 1. Concretely, for each lexical category, we use an LLM to embed all the contained words and then average the resulting embeddings to get a set $C$ of $k$ centroids: $C = \{c_1, c_2, \ldots, c_k\}$. See Appendix A.3 for more details. Then, a proposed expert feature $\hat{g} \in \{0, 1\}^d$ indicates whether or not each of the $d$ words $w_1, w_2, ..., w_d \in x$ are included in the feature, and the expert alignment score for the proposed feature $\hat{g}$ can be computed as follows:

$$\text{EXPERTALIGN}(\hat{g}, x) = \max_{c \in C} \frac{1}{|\hat{g}|} \sum_{i=1}^{d} \hat{g}_i \cdot \cos(\text{embedding}(w_i), c) \qquad (6)$$

### 4.4 Emotion Dataset

**Motivation.** Emotion classification involves inferring the emotion (e.g., Joy, Anger, etc.) reflected in a piece of text. Researchers study emotion to build systems that can understand emotion and thus adapt accordingly when interacting with human users. For extracted features to be useful for such systems, they must be relevant to emotion. For example, a word like "puppy" may be used more frequently in comments labeled with Joy vs. other emotions; therefore, it may be extracted as a relevant feature for the Joy class. However, this is a spurious correlation — emotional expression is not necessarily tied to a subject, and comments containing "puppy" may also be angry or sad.

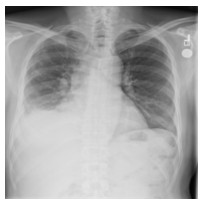 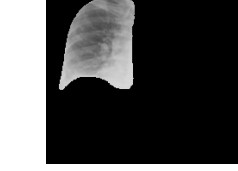 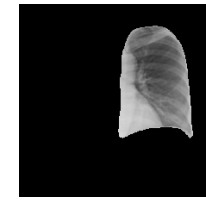

(a) Full image        (b) Right lung        (c) Left lung

Figure 5: An example with expert features for Chest X-Ray dataset. (a) The full X-ray image where the following pathologies are present: effusion, infiltration, and pneumothorax; (b-c) Expert-interpretable anatomical structures of the left and right lungs.

**Problem Setup.** The GoEmotions dataset from Demszky et al. (2020) contains 58,000 English Reddit comments labeled for 27 emotion categories, or "neutral" if no emotion is applicable. The input is a text utterance of 1-2 sentences extracted from Reddit comments, and the output is a binary label for each of the 27 emotion categories. The dataset contains train/validation/test splits of sizes 43,400/5,430/5,430, respectively.

**Expert Features.** Example expert features are shown in Table 1. To measure how emotion-related a feature is, we use the circumplex model of affect (Russell, 1980). The circumplex model assumes that all emotions can be projected onto the 2D unit circle with respect to two independent dimensions – *arousal* (the magnitude of intensity or activation) and *valence* (how negative or positive). By projecting features onto the unit circle, we can quantify emotional relations. In particular, we calculate the following two attributes of the features with a group: (1) their emotional *signal*, i.e., mean distance to the circumplex and (2) their emotional *relatedness*, i.e., mean pairwise distance within the circumplex. We then calculate the following: $\text{Signal}(\hat{g}, x)$, which measures the average Euclidean distance to the circumplex for every projected feature in $\hat{g}$, and $\text{Relatedness}(\hat{g}, x)$, which measures the average pairwise distance between every projected feature in $\hat{g}$ (details in Appendix A.4). For an extracted feature $\hat{g}$, the expert alignment score can then be computed by:

$$\text{EXPERTALIGN}(\hat{g}, x) = \tanh(\exp[-\text{Signal}(\hat{g}, x) \cdot \text{Relatedness}(\hat{g}, x)]) \tag{7}$$

### 4.5 Chest X-Ray Dataset

**Motivation.** Chest X-ray imaging is a common procedure for diagnosing conditions such as atelectasis, cardiomegaly, and effusion, among others. Although radiologists are skilled at analyzing such images, modern machine learning models are increasingly competitive in diagnostic performance (Ahmad, 2021). Therefore, ML models may prove useful in assisting radiologists in making diagnoses. However, in the absence of an explanation, radiologists may only trust the model output if it matches their own predictions. Moreover, inaccurate AI assistants are shown to negatively affect diagnostic performance (Yu et al., 2024). To address this problem, explainability could be employed as a safeguard to help radiologists decide whether or not to trust the model. As such, it is important for machine learning models to provide explanations for their diagnoses.

**Problem Setup.** We use the NIH-Google dataset (Majkowska et al., 2020) available from the TorchXRayVision library (Cohen et al., 2022). This is a relabeling of the NIH

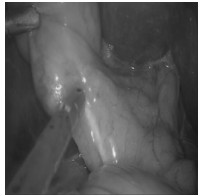
(a) Full image

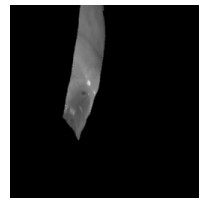
(b) Safe region

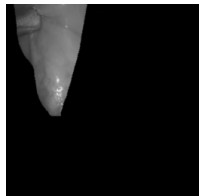
(c) Gallbladder

Figure 6:    An example with expert features of Laparoscopic Cholecystectomy Surgery Dataset: (a) The view of the surgeon sees; (b) The safe region for operations; (c) The gallbladder, a key anatomical structure for the critical view of safety.

ChestX-ray14 dataset (Wang et al., 2017) which improved the quality of the original labels. It contains 28,868 chest X-ray images labeled for 14 common pathology categories: atelectasis, calcification, cardiomegaly, etc. We randomly partition the dataset into train/test splits of sizes 23,094/5,774, respectively. The task is a multi-label classification problem for identifying the presence of each pathology.

**Expert Features.** Radiology reports commonly refer to anatomical structures (e.g., spine, lungs), which allows radiologists to perform and communicate accurate diagnoses to patients. We provide these expert-interpretable features in the form of anatomical structure segmentations. However, because we could not find datasets with both pathology labels and anatomical segmentations, we used a pre-trained model from TorchXRayVision to generate the structure labelings for each image. We use explicit expert alignment as described in Equation 3 to compute alignment of an extracted feature $\hat{g}$ and the 14 predicted anatomical structure segments, including the left clavicle, heart, etc. Details of the Chest X-Ray dataset can be found in Appendix A.5.

### 4.6 Laparoscopic Cholecystectomy Surgery Dataset

**Motivation.** Laparoscopic cholecystectomy (gallbladder removal) is one of the most common elective abdominal surgeries performed in the US, with over 750,000 operations annually (Stinton and Shaffer, 2012). A common complication of laparoscopic surgery is bile duct injury, which is associated with an 8-fold increase in mortality (Michael Brunt et al., 2020) and accounts for more than $1B in US healthcare annual spending (Berci et al., 2013). Notably, 97% of such complications result from human visualization errors (Way et al., 2003). The surgery site commonly contains obstructing tissues, inflammation, and other patient-specific artifacts — all of which may prevent the surgeon from getting a perfect view. Consequently, there is growing interest in harnessing advanced vision models to help surgeons distinguish safe and risky areas for operation. However, experienced surgeons rarely trust model outputs due to their opaque nature, while inexperienced surgeons might overly rely on model predictions. Therefore, any safe and useful machine learning model must be able to provide explanations that align with surgeons' expectations.

**Problem Setup.** The task is to identify the safe and unsafe regions for incision. We use the open-source subset of the data from (Madani et al., 2022), wherein the authors enlist surgeons to annotate surgery video data from the M2CAI16 workflow challenge (Stauder et al., 2016) and Cholec80 (Twinanda et al., 2016) datasets. This dataset consists of 1015

| | | Vision | | | Time Series | | Language | | |
|---|---|---|---|---|---|---|---|---|---|
| | **Method** | **Cholec** | **ChestX** | **MassMaps** | **Method** | **Supernova** | **Method** | **Politeness** | **Emotion** |
| *Domain-specific* | Identity | 0.4648 | 0.2154 | 0.5483 | Identity | 0.0152 | Identity | 0.6070 | 0.0103 |
| | Random | 0.1084 | 0.0427 | 0.5505 | Random | 0.0358 | Random | 0.6478 | 0.0303 |
| | Patch | 0.0327 | 0.0999 | 0.5555 | Slice 5 | 0.0337 | Words | 0.6851 | 0.1182 |
| | Quickshift | 0.2664 | 0.3419 | 0.5492 | Slice 10 | 0.0555 | Phrases | 0.6351 | 0.0198 |
| | Watershed | 0.2806 | 0.1452 | 0.5590 | Slice 15 | 0.0554 | Sentences | 0.6109 | 0.0120 |
| | SAM | 0.3642 | 0.3151 | 0.5521 | | | | | |
| | CRAFT | 0.0278 | 0.1175 | 0.3996 | | | | | |
| *Domain-agnostic* | Clustering | 0.2839 | 0.2627 | 0.5515 | Clustering | 0.2622 | Clustering | 0.6680 | 0.0912 |
| | Archipelago | 0.3271 | 0.2148 | 0.5542 | Archipelago | 0.2574 | Archipelago | 0.6773 | 0.0527 |

Table 2: Baselines scores of different FIX settings. We report the mean score and give a more comprehensive table in Appendix C. We describe baseline implementations in Section 5. One thing to note is that FIXSCORE is not comparable for different tasks (e.g. between Mass Maps and Supernova) as the data and specific expert alignment metrics are different for different tasks.

annotated images that are randomly split by video sources, with train/test splits of sizes 785/230, respectively.

**Expert Features.** In cholecystectomy, it is a common practice for surgeons to identify the *critical view of safety* before performing any irreversible operations (Strasberg and Brunt, 2010; Hashimoto et al., 2019). This view identifies the location of vital organs and structures that inform the safe region of operation and is incidentally what surgeons often need as part of an explanation. We provide these expert-interpretable labels in the form of organ segmentations (liver, gallbladder, hepatocystic triangle). We use explicit expert alignment as described in Equation 3 to compute alignment of an extracted feature $\hat{g}$ and the surgeon-annotated organ labels taken from Madani et al. (2022). Details of the Cholecystectomy dataset can be found in Appendix A.6.

## 5 Baseline Algorithms & Discussion

We evaluate standard techniques widely used within the vision, text, and time series domains to create higher-level features. We provide a brief summary below, with additional details in Appendix C.

**Domain-specific Baselines.** We consider the following domain-centric baselines, which are standard in the literature for the respective domains. *(Image)* For image data, we consider three segmentation methods (Kim et al., 2024). Patches (Dosovitskiy et al., 2021) divides the image into grids where each cell is the same size. Quickshift (Grady, 2006) connects similar neighboring pixels into a common superpixel. Watershed (Levner and Zhang, 2007) simulates flooding on a topographic surface. Segment Anything Model (SAM) (Kirillov et al., 2023) is a large foundation model for generating image segmentations. CRAFT (Fel et al., 2023) generates concept attribution maps. *(Time-series)* For time series data, we take equal size slices of the data across time as patches (Schlegel et al., 2021). We use different slice sizes to see how they impact multiple baselines. We take various slice sizes, such as 5, 10, and 15, separately to evaluate the results of multiple baselines. *(Text)* For text data, we present three baselines for extracting features (Rychener et al., 2022). At the finest granularity, we

treat each word as a feature. The second baseline considers each phrase as a feature. Phrases are comprised of groups of words that are separated by some punctuation in the original text. At the coarsest granularity, we treat each sentence as a feature.

**Domain-agnostic Baselines.** We additionally consider the following domain-agnostic baselines for feature extraction. *(Identity)* We combine all elements into one single group. *(Random)* We select features at random, up to the maximum baseline results for the group. The group maximum is calculated as: (group maximum) $\approx$ (scaling factor) $\times$ (number of expert features). The size of the distinct expert feature varies depending on the setting, and further details for each setting can be found in Appendix C. We use a scaling factor of about 1.5 to allow for flexibility. *(Clustering)* For images, we first use Quickshift to generate segments and then pass each segment through a feature extractor (ResNet-18 by default). For time series, we use raw features from each time segment. We then apply K-means clustering on the extracted/raw features to relabel and merge segments. For text, we use BERTopic (Grootendorst, 2022) to obtain the clusters. *(Archipelago)* We adapt the implementation of Archipelago (Tsang et al., 2020) to use ResNet-18 with quickshift for feature extraction.

**Results and Discussions.** We show results on the baselines in Table 2. For image datasets, Quickshift has the best performance compared to Patch and Watershed on both the Cholecystectomy dataset and the Chest X-ray dataset, since they have natural images. All baselines perform similarly for the Mass Maps dataset. That the range of mass maps is different from other tasks is potentially because they are not natural images, but rather similar to topographic surfaces, and also the implicit ground truth expert features do not have full coverage. For the Supernova time-series dataset, larger slices score yield higher expert alignment scores. For both Multilingual Politeness and Emotion datasets, individual words appear to be the most expert-aligned features. Generally, however, we see that the domain-agnostic neural baselines tend to also perform better than or close to the best domain-centric baseline. The main benefit of using a neural approach is that it can more easily automatically discover relevant features.

## 6 Conclusion

We propose FIX, a curated benchmark of datasets with evaluation metrics for extracting expert features in diverse real-world settings. Our benchmark addresses a gap in the literature by providing researchers with an environment to study and automatically extract interpretable features for experts, designed by experts.

**Limitations and Future Work.** The FIX benchmark is not an exhaustive specification of all expert features, and may fail to capture others types. The ones we included are generally non-controversial and well-accepted by the domain's expert community, but we can foresee that there are cases where this may not be true. Dealing with potential conflicting expert opinions may need a more nuanced approach, which is left for future work to address. Furthermore, although we cover cosmology, psychology, and medicine domains in this work, the metrics for these domains may not be appropriate for all settings. We encourage prospective users to consider and implement metrics most appropriate to their particular settings. Future work includes the development of new, general purpose techniques that can extract expert features from data and models without supervision. Additionally, future work

could also include training machine learning models on just the features that are deemed to be aligned with domain experts and reporting the accuracy of the trained models.

## Broader Impact and Ethics Statement

The goal of the FIX benchmark is to enable researchers and practitioners to develop more transparent machine learning systems that are applicable in real-world problems. However, because our datasets contain text scraped from Internet forums, as well as visuals of human anatomy, it is possible that some contents may be considered objectionable. It is possible that such objectionable content may be misused, but we do not believe that our datasets would be of particular interest to malicious users because dedicated natural-language toxicity and more graphic medical datasets exist.

## Acknowledgment

This research was partially supported by a gift from AWS AI to Penn Engineering's ASSET Center for Trustworthy AI, by ASSET Center Seed Grant, ARPA-H program on Safe and Explainable AI under the award D24AC00253-00, by NSF award CCF 2442421, and by funding from the Defense Advanced Research Projects Agency's (DARPA) SciFy program (Agreement No. HR00112520300). The views expressed are those of the author and do not reflect the official policy or position of the Department of Defense or the U.S. Government.

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

# Appendix A. Dataset Details

All datasets and their respective Croissant metadata records and licenses are available on HuggingFace at the following links.

- **Mass Maps**:
    https://huggingface.co/datasets/BrachioLab/massmaps-cosmogrid-100k
- **Supernova**:
    https://huggingface.co/datasets/BrachioLab/supernova-timeseries
- **Multilingual Politeness**:
    https://huggingface.co/datasets/BrachioLab/multilingual_politeness
- **Emotion**:
    https://huggingface.co/datasets/BrachioLab/emotion
- **Chest X-Ray**:
    https://huggingface.co/datasets/BrachioLab/chestx
- **Laparoscopic Cholecystectomy Surgery**:
    https://huggingface.co/datasets/BrachioLab/cholec

## A.1 Mass Maps Dataset

**Problem Setup.**  We randomly split the data to consist of 90,000 train and 10,000 validation maps and maintain the original 10,000 test maps. We follow the post-processing procedure in Jeffrey et al. (2021); You et al. (2023) for low-noise maps. Following previous works (Ribli et al., 2019; Matilla et al., 2020; Fluri et al., 2022; You et al., 2023), we use a CNN-based model for predicting $\Omega_m$ and $\sigma_8$.

**Metric.**  Let $x \in \mathbb{R}^d$ be the input mass map with $d = H \times W$ pixels, and $g \in \{0,1\}^d$ be a boolean mask $g$ that describes which pixels belong to the group, where $g_i = 1$ if the $i$th pixel belongs to the group, and 0 otherwise.

We can compute the purity score of each group to void and cluster. We say a pixel is a void (underdensed) pixel if its intensity is below 0, and a cluster (overdensed) pixel if its intensity is above $3\sigma(x)$, following previous works (Matilla et al., 2020; You et al., 2023). We first compute the proportion of void pixels and cluster pixels in feature $g$

$$P_v(g, x) = \frac{\sum_{i=1}^{d} \mathbb{1}[g_i x_i < 0]}{g^\intercal \mathbf{1}}, \qquad P_c(g, x) = \frac{\sum_{i=1}^{d} \mathbb{1}[g_i x_i > 3\sigma(x)]}{g^\intercal \mathbf{1}} \tag{8}$$

where $\mathbf{1} \in 1^d$ is the identity matrix, the numerators count the number of underdensed or overdensed pixels, and $g^\intercal \mathbf{1}$ is the number of pixels in the feature. In practice, we add a small $\epsilon = 10^{-6}$ to $P_v$ and $P_c$ and renormalize them, to avoid taking the log of 0 later. Next, we compute the proportion of pixels that are void or cluster, only among the void/cluster pixels:

$$P'_v(g, x) = \frac{P_v(g, x)}{P_v(g, x) + P_c(g, x)}, \qquad P'_c(g, x) = \frac{P_c(g, x)}{P_v(g, x) + P_c(g, x)} \tag{9}$$

Then, we compute the EXPERTALIGN score for the predicted feature $\hat{g}$ by computing the void/cluster-only entropy reversed and scaled to $[0, 1]$, weighted by the percentage of void/cluster pixels among all pixels.

$$\text{Purity}_{vc}(\hat{g}, x) = \frac{1}{2}(2 + P'_v(\hat{g}, x) \log_2 P'_v(\hat{g}, x) + P'_c(\hat{g}, x) \log_2 P'_c(\hat{g}, x)) \tag{10}$$

where $-(P'_v(\hat{g}, x) \log_2 P'_v(\hat{g}, x) + P'_c(\hat{g}, x) \log_2 P'_c(\hat{g}, x))$ is the entropy computed only on void and cluster pixels, a close to 0 score indicating that the interpretable portion of the feature is mostly void or cluster. $\text{Purity}_{vc}(\hat{g}, x)$ is 0 if among the pixels in the proposed feature that are either void or cluster pixels, half are void and half are cluster pixels, and 1 if all are void or all are cluster pixels, regardless of how many other pixels there are in the proposed feature.

We also have the ratio

$$\text{Ratio}_{vc}(\hat{g}, x) = (P_v(\hat{g}, x) + P_c(\hat{g}, x)) \tag{11}$$

which is the total proportion of the feature that is any interpretable feature type at all.

We then have our EXPERTALIGN for Mass Maps:

$$\text{EXPERTALIGN}(\hat{g}, x) = \text{Purity}(\hat{g}, x) \cdot \text{Ratio}(\hat{g}, x) \tag{12}$$

which is then 0 when all the pixels in the feature are neither void or cluster, and 1 if all pixels are void pixels or all pixels are cluster pixels, and somewhere in the middle if most pixels are void or cluster pixels but there is a mix between both.

## A.2 Supernova Dataset

**Problem Setup.** We extracted data from the PLAsTiCC Astronomical Classification challenge (Team et al., 2018). [*] PLAsTiCC dataset was designed to replicate a selection of observed objects with type information typically used to train a machine learning classifier. The challenge aims to categorize a realistic simulation of all LSST observations that are dimmer and more distorted than those in the training set. The dataset contains 15 classes, with 14 of them present in the training sample. The remaining class is intended to encompass intriguing objects that are theorized to exist but have not yet been observed.

In our dataset, we split the original training set into 90/10 training/validation, and the original test set was uploaded unchanged. We made these sets balanced for each class. The class includes objects such as tidal disruption event (TDE), peculiar type Ia supernova (SNIax), type Ibc supernova (SNIbc), and kilonova (KN). The dataset contains four columns: observation times (modified Julian days, MJD), wavelength (filter), flux values, and flux error. Spectroscopy measures the flux with respect to wavelength, similar to using a prism to split light into different colors.

Due to the expected high volume of data from upcoming sky surveys, it is not possible to obtain spectroscopic observations for every object. However, these observations are crucial for us. Therefore, we use an approach to capture images of objects through different filters, where each filter selects light within a specific broad wavelength range. The supernova dataset includes 7 different wavelengths that are used. The flux values and errors are recorded at specific time intervals for each wavelength. These values are utilized to predict the class that this data should be classified into.

**Metric.** We use the following expert alignment metric to measure if a group of features is interpretable:

$$\text{EXPERTALIGN}(\hat{g}, x) = \max_{w \in W} \text{LinearConsistency}(\hat{g}, x_w) \tag{13}$$

---

[*] `https://www.kaggle.com/c/PLAsTiCC-2018`

where $W$ is the set of unique wavelength, $\hat{g}$ is the feature group, and $x_w$ is the subset of $x$ within wavelength $w$. In the supernova setting, there are three parameters: $\epsilon$, the parameter for how much standard deviation $\sigma$ is allowed, window size $\lambda$ and the step size $\tau$. Therefore, we formulate the LinearConsistency function as follows:

$$\text{LinearConsistency}(\hat{g}, x_w) = p(\hat{g}, x_w) \cdot d(\hat{g}, x_w) \tag{14}$$

$p(\hat{g}, x_w)$ is the percentage of data points that display linear consistency, penalized by $d(\hat{g}, x_w)$, which is the percentage of time steps containing data points.

Let $\beta(x, y) = \arg\min_\beta (X^T\beta - y)^2$, where $X = \begin{bmatrix} x & 1 \end{bmatrix}$ and $\beta = \begin{bmatrix} \beta_1 & \beta_0 \end{bmatrix}$. Here, $\beta_1$ is the slope and $\beta_0$ is the intercept. $M$ is the number of data points in $x_w$, and $\hat{y}_{w,i} = x_{w,i} \cdot \beta$. Then, we have

$$\text{p}(\hat{g}, x_w) = \frac{1}{M} \sum_{i=1}^{M} \mathbb{1}[\hat{y}_{w,i} \in [y_{w,i} - \epsilon \cdot \omega_{w,i}, y_{w,i} + \epsilon \cdot \omega_{w,i}]] \tag{15}$$

Let $t_1, ..., t_N$ be time steps at step size intervals. Then $t_i = t_{start} + i * \tau$, and $N$ is the number of time steps. We also have

$$\text{d}(\hat{g}, x_w) = \frac{1}{N} \sum_{i=1}^{N} \mathbb{1}[\exists_i : x_{w,i} \in [t_i, t_i + \lambda]] \tag{16}$$

A higher $\text{EXPERTALIGN}(\hat{g}, x) \in [0, 1]$ value means the flux slope at each wavelength is consistently linear and there are not many time intervals without data.

## A.3 Multilingual Politeness Dataset

**Problem Setup.** This politeness dataset from Havaldar et al. (2023b) is intended for politeness classification, and would likely be solved via a fine-tuned multilingual LLM. Namely, this would be a regression task, using a trained LLM to output the politeness level of a given conversation snippet as a real number ranging from -2 to 2.

The dataset is accompanied by a theory-grounded politeness lexica. Such lexica built with domain expert input have been promising for explaining style (Danescu-Niculescu-Mizil et al., 2013), culture (Havaldar et al., 2024), and other such complex multilingual constructs.

**Metric.** Assume a theory-grounded Lexica $L$ with $k$ categories: $L = \ell_1, \ell_2, ...\ell_k$, where each set $\ell_i \subseteq \mathcal{W}$, where $\mathcal{W}$ is the set of all words. For each category, we use an LLM to embed all the contained words and then average the resulting embeddings, to get a set $C$ of $k$ centroids: $C = c_1, c_2, ...c_k$. We define this formally as:

$$C : \left\{ \frac{1}{|\ell_i|} \sum_{w \in l_i} \text{embedding}(w) \text{ for all } i \in [1, k] \right\} \tag{17}$$

For a group $\hat{g}$ containing words $w_1, w_2, ...$, the group-level expert alignment score can be computed as follows:

$$\text{EXPERTALIGN}(\hat{g}, x) = \max_{c \in C} \frac{1}{|\hat{g}|} \sum_{w \in \hat{g}} \cos(\text{embedding}(w), c) \tag{18}$$

Note that each language has a different theory-grounded lexicon, so we calculate a unique domain alignment score for each language.

### A.4 Emotion Dataset

**Problem Setup.** This dataset is intended for emotion classification and is currently solved with a fine-tuned LLM (Demszky et al., 2020). Namely, this is a classification task where an LLM is trained to select some subset of 28 emotions (including neutrality) given a 1-2 sentence Reddit comment.

| Axis Anchor | Russell Emotions |
|---|---|
| Positive valence (PV) | Happy, Pleased, Delighted, Excited, Satisfied |
| Negative valence (NV) | Miserable, Frustrated, Sad, Depressed, Afraid |
| High arousal (HA) | Astonished, Alarmed, Angry, Afraid, Excited |
| Low arousal (LA) | Tired, Sleepy, Calm, Satisfied, Depressed |

Table 3: Emotions used to define the valence and arousal axis anchors for projection into the Valence-Arousal plane. We select the 5 emotions from the circumplex closest to each axis point.

**Projection onto the Circumplex.** To define the valence and arousal axes, we first generate four axis-defining points by averaging the contextualized embeddings ("I feel [emotion]") of the emotions listed in Table 3. This gives us four vectors in embedding space – positive valence ($\vec{v}_{\text{pos}}$), negative valence($\vec{v}_{\text{neg}}$), high arousal($\vec{a}_{\text{high}}$), and low arousal($\vec{a}_{\text{low}}$). We mathematically describe our projection function below:

1. We define the valence axis, $V$, as $\vec{v}_{\text{pos}} - \vec{v}_{\text{neg}}$ and the arousal axis, $A$, as $\vec{a}_{\text{high}} - \vec{a}_{\text{low}}$. We then normalize $V$ and $A$ and calculate the origin as the midpoints of these axes: $(\vec{v}_{\text{middle}}, \vec{a}_{\text{middle}})$.

2. We then scale the axes so $\vec{v}_{\text{pos}}$, $\vec{v}_{\text{neg}}$, $\vec{a}_{\text{high}}$, and $\vec{a}_{\text{low}}$ anchor to $(1, 0)$, $(-1, 0)$, $(0, 1)$, and $(0, -1)$ respectively. This enforces the circumplex to be a unit circle in the valence-arousal plane.

3. We compute the angle $\theta$ between the valence-arousal axes by solving $\cos\theta = \frac{V \cdot A}{\|V\| \cdot \|A\|}$

4. For each embedding vector $\vec{x}$ in the set $\{x_i\}_{i=1}^n$ we want to project into our defined plane, we compute the valence and arousal components for $x_i$ as follows:
$x_i^v = (x_i - \vec{v}_{\text{middle}}) \cdot \vec{V}$
$x_i^a = (x_i - \vec{a}_{\text{middle}}) \cdot \vec{A}$.

5. We calculate the x and y coordinates to plot, enforcing orthogonality between the axes:
$\tilde{x}_i^v = x_i^v - x_i^a \cdot \cos\theta$
$\tilde{x}_i^a = x_i^a - x_i^v \cdot \cos\theta$

6. Finally, we plot $(\tilde{x}_i^v, \tilde{x}_i^v)$ in the Valence-Arousal plane. We then calculate the shortest distance from $(\tilde{x}_i^v, \tilde{x}_i^v)$ to the circumplex unit circle.

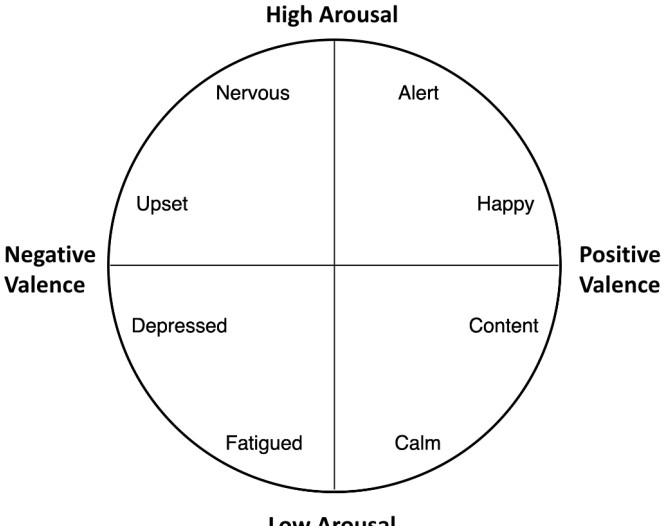

Figure 7: The circumplex model of affect Russell (1980).

**Metric.** We calculate the following two values for a proposed feature $\hat{g}$ containing words $w_1, w_2, ...$, where $n$ is the number of words in $\hat{g}$:

$$\text{Signal}(\hat{g}) = \frac{1}{n} \sum_{w \in \hat{g}} \left| \|\text{Proj}(w)\|_2 - 1 \right| \tag{19}$$

$$\text{Relatedness}(\hat{g}) = \frac{1}{n^2} \sum_{i}^{n} \sum_{j}^{n} \|\text{Proj}(w_i) - \text{Proj}(w_j)\|_2 \tag{20}$$

where $\text{Signal}(\hat{g}, x)$ measures the average Euclidean distance to the circumplex for every projected feature in $\hat{g}$, and $\text{Relatedness}(\hat{g}, x)$ measures the average pairwise distance between every projected feature in $\hat{g}$. We formalize the expert alignment metric as follows. For a group $\hat{g}$, the expert alignment score can be computed by:

$$\text{EXPERTALIGN}(\hat{g}, x) = \tanh(\exp[-\text{Signal}(\hat{g}, x) \cdot \text{Relatedness}(\hat{g}, x)]) \tag{21}$$

### A.5 Chest X-Ray Dataset

We used datasets and pretrained models from TorchXRayVision (Cohen et al., 2022).[*] In particular, we use the NIH-Google dataset (Majkowska et al., 2020), which is a relabeling of the NIH ChestX-ray14 dataset (Wang et al., 2017). This dataset contains 28,868 chest X-ray images labeled for 14 common pathology categories, with a train/test split of 23,094 and 5,774. We additionally used a pre-trained structure segmentation model to produce 14 segmentations. The task is a multi-label classification problem for identifying the presence of each pathology. The 14 pathologies are:

---

[*] `https://github.com/mlmed/torchxrayvision`

Atelectasis, Cardiomegaly, Consolidation, Edema, Effusion, Emphysema, Fibrosis, Hernia, Infiltration, Mass, Nodule, Pleural Thickening, Pneumonia, Pneumothorax

The 14 anatomical structures are:

Left Clavicle, Right Clavicle, Left Scapula, Right Scapula, Left Lung, Right Lung, Left Hilus Pulmonis, Right Hilus Pulmonis, Heart, Aorta, Facies Diaphragmatica, Mediastinum, Weasand, Spine

### A.6 Laparoscopic Cholecystectomy Surgery Dataset

We use the open-source subset of the data from (Madani et al., 2022), which consists of surgeon-annotated video data taken from the M2CAI16 workflow challenge (Stauder et al., 2016) and Cholec80 (Twinanda et al., 2016) datasets. The task is to identify the safe/unsafe regions of where to operate. Specifically, each pixel of the image has one of three labels: background, safe, or unsafe. The expert labels provide each pixel with one of four labels: background, liver, gallbladder, and hepatocystic triangle.

## Appendix B. Interpretable Feature Extraction Details

Figure 8 illustrates a graphical model representing the Interpretable Feature Extraction pipeline for a given FIX dataset.

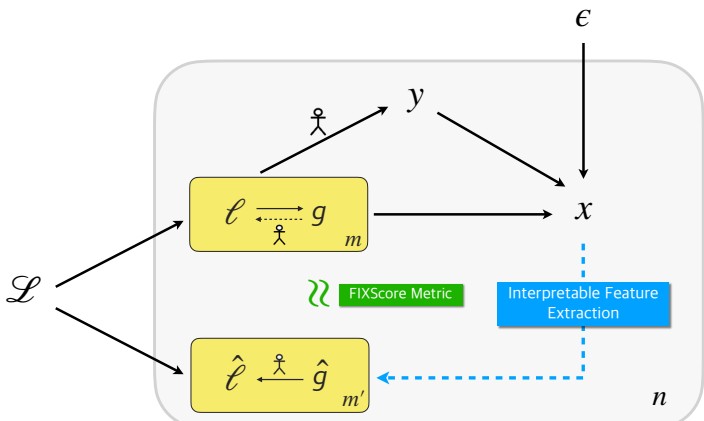

Figure 8: We illustrate a graphical model representing the Interpretable Feature Extraction pipeline for a given FIX dataset, with FIXScore metric in its general form. There are $m$ true feature groups $g$ and $m$ latent features $\ell$, and $m'$ proposed feature groups $\hat{g}$ and $m'$ proposed latent features $\hat{\ell}$. $m$ does not have to equal $m'$. Moreover, $n$ indicates the number of examples in the dataset. The person figure on near the closest arrow indicates that a domain expert would be able to infer the variable on the right-hand side of the arrow from the variable on the left-hand side arrow. In addition, $\epsilon$ is included to account for noise.

## Appendix C. Baselines Details

The FIX benchmark is publicly available at: `https://brachiolab.github.io/fix/`

**Bootstrapping.** For each setting's baselines experiments, we use a bootstrapping method (with replacement) to estimate the standard deviation of the sample means of FIXScore.

**Group Maximum.** For the number of groups, we take the scaling factor multiplied by the size of the distinct expert feature, which differs for each setting. The scaling factor we choose across all setting is 1.5 (and round up to the next nice whole number).

In the case of a supernova setting, we consider a distinct expert feature size of 6. This is because the maximum number of distinct expert features we can obtain is 6, given that there are a maximum of 3 humps in the time series dataset. For each hump, there are both peaks and troughs, leading to a potential maximum of 6 distinct expert features.

For the multilingual politeness setting, the group maximum would be 40, which is the total number of lexical categories, 26, with the scaling factor multiplied in to give some flexibility.

For the emotion setting, the group maximum would be , which is the total number of lexical categories, 26, with the scaling factor multiplied in to give some flexibility.

For mass maps, the group maximum would be 25. We compute the maximum number of local maximums 7 on mass maps blurred with $\sigma = 3$ and local minimums 7 on mass maps blurred with $\sigma = 5$, which sums up to be 14. We can then multiply with the scaling factor to give some flexibility and then we round up to 25.

**Baseline Parameters.** For mass maps, we use the following parameters for baselines. For patch, we use $8 \times 8$ grid. For QuickShift, we use kernel size 5, max dist 10, and sigma 0.2. For watershed, we use min dist 10, compactness 0. For SAM, we use 'vit_h'. For Archipelago, we use the same Quickshift parameters for the Quickshift segmenter.

**Baseline Results.** We report the full baseline results with standard deviations in Table 4.

## Appendix D. Representative Examples of Extracted Features.

Here, we include representative examples of features extracted by existing baseline methods, along with commentary on how they differ from expert-aligned features.

### D.1 Mass Maps Dataset

**Example Features.** As MassMaps does not have annotated expert features, we only show example of generated features with corresponding percent void and cluster and alignment scores in Figure 9. We can see that the 6th feature (3rd image on the second row) achieves the highest alignment score with a large percentage of void (86.3%) and a very small percent of cluster (0.8%), while the 5th features (2nd image on the second row) has the lowest alignment of (57.3%), as it is not fully aligned to either void or cluster.

### D.2 Supernova Dataset

See Figure 10.

| | Method | Cholecystectomy | Chest X-ray | Mass Maps |
|---|---|---|---|---|
| *Image* | Identity | 0.4648 ± 0.0045 | 0.2154 ± 0.0027 | 0.5483 ± 0.0015 |
| | Random | 0.1084 ± 0.0004 | 0.0427 ± 0.0001 | 0.5505 ± 0.0014 |
| | Patch | 0.0327 ± 0.0001 | 0.0999 ± 0.0008 | 0.5555 ± 0.0013 |
| | Quickshift | 0.2664 ± 0.0036 | 0.3419 ± 0.0025 | 0.5492 ± 0.0009 |
| | Watershed | 0.2806 ± 0.0049 | 0.1452 ± 0.0017 | 0.5590 ± 0.0017 |
| | SAM | 0.3642 ± 0.0092 | 0.3151 ± 0.0064 | 0.5521 ± 0.0009 |
| | CRAFT | 0.0278 ± 0.0003 | 0.1175 ± 0.0011 | 0.3996 ± 0.0023 |
| *Domain-Agnostic* | Clustering | 0.2839 ± 0.0024 | 0.2627 ± 0.0039 | 0.5515 ± 0.0014 |
| | Archipelago | 0.3271 ± 0.0076 | 0.2148 ± 0.0009 | 0.5542 ± 0.0014 |
| | | **Supernova** | | |
| *Time Series* | Identity | 0.0152 ± 0.0011 | | |
| | Random | 0.0358 ± 0.0021 | | |
| | Slice 5 | 0.0337 ± 0.0015 | | |
| | Slice 10 | 0.0555 ± 0.0044 | | |
| | Slice 15 | 0.0554 ± 0.0032 | | |
| *Domain-Agnostic* | Clustering | 0.2622 ± 0.0037 | | |
| | Archipelago | 0.2574 ± 0.0082 | | |
| | | **Multilingual Politeness** | **Emotion** | |
| *Text* | Identity | 0.6070 ± 0.0015 | 0.0103 ± 0.0001 | |
| | Random | 0.6478 ± 0.0012 | 0.0303 ± 0.0004 | |
| | Words | 0.6851 ± 0.0010 | 0.1182 ± 0.0003 | |
| | Phrases | 0.6351 ± 0.0010 | 0.0198 ± 0.0003 | |
| | Sentences | 0.6109 ± 0.0006 | 0.0120 ± 0.0002 | |
| *Domain-Agnostic* | Clustering | 0.6680 ± 0.0048 | 0.0912 ± 0.0005 | |
| | Archipelago | 0.6773 ± 0.0006 | 0.0527 ± 0.0008 | |

Table 4: Baselines of different FIX settings. We report the mean FIXSCORE for all examples in each setting, with standard deviations.

### D.3 Multilingual Politeness Dataset

**Example Features.** Since the multilingual politeness dataset does not have annotated expert features, we use semantic similarity with the politeness lexica in Havaldar et al. (2023a), adapted from the Stanford Politeness Lexicon (Danescu-Niculescu-Mizil et al., 2013).

A feature for the multilingual politeness dataset is a single word. We choose to not further break down words into tokens, as it is unclear what the cosine similarity between a token and a word in a lexicon would mean. In this vein, feature groups are a collection of words in the input that need not appear consecutively.

**Expert Features.** An expert feature is a lexical category from the Stanford Politeness Lexicon (Danescu-Niculescu-Mizil et al., 2013). Such categories include apology words, greetings, positive sentiment words, etc., where each category is either an indicator of politeness or an indicator of rudeness. see Table 5 for examples of such expert features.

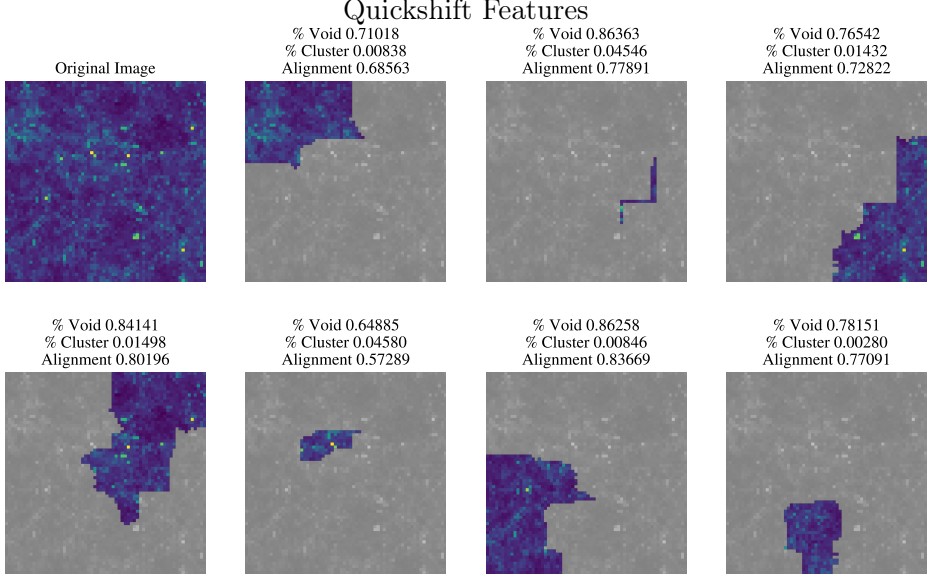

Figure 9: MassMaps features from quickshift with void, cluster, and expert alignment scores.

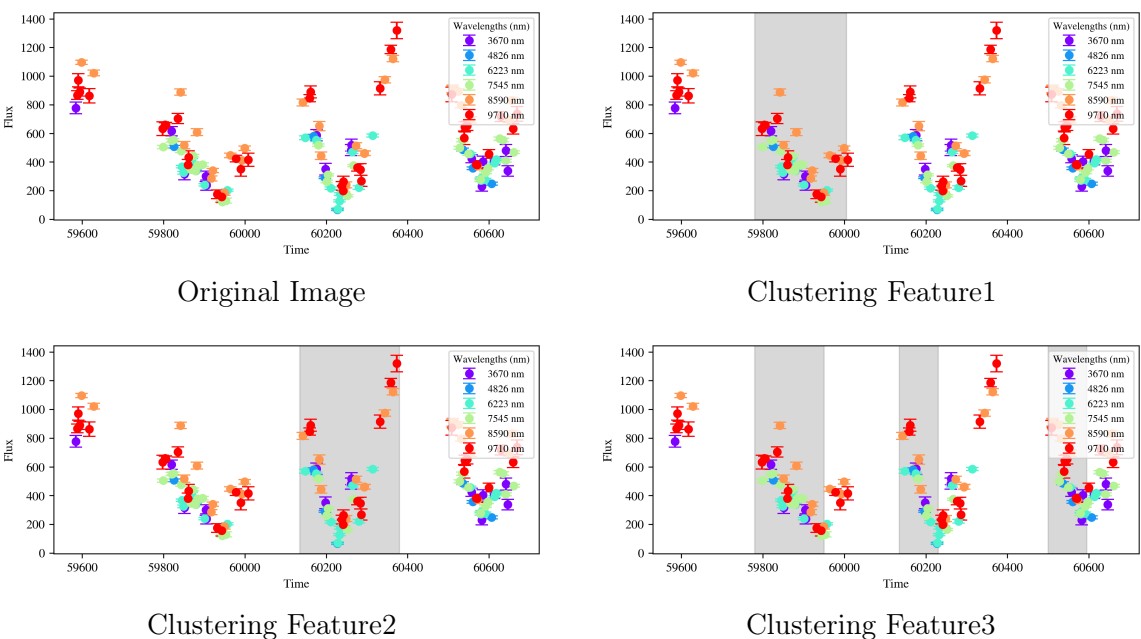

Figure 10: Supernova features from clustering.

## D.4 Emotion Dataset

**Example Features.** The emotion dataset also does not have annotated expert features, so we use valence and arousal signal (Russell, 1980).

| Input | Example Feature | Expert Feature |
|---|---|---|
| I was running my spellchecker and totally didn't realize that this was a vandalized page. Please accept my apology. I will spellcheck a little slower next time. | "my" | First-person pronouns: *I, my, mine, etc.* |
| | "vandalized" | Negative sentiment: *bad, ugly, terrorized, etc.* |
| | "apology" | Apologizing: *sorry, apology, my bad, etc.* |

Table 5: Example features and corresponding expert features for the multilingual politeness dataset.

| Input | Example Feature | Expert Feature |
|---|---|---|
| This was potentially the most dangerous stunt I have ever seen someone do. One minor mistake and you die. | "dangerous" | Low Valence: *death, horrible, scary, etc.* |
| | "minor" | Low Arousal: *calm, tired, unexciting, etc.* |
| | "stunt" | High Arousal: *furious, excited, surprised, etc.* |

Table 6: Example features and corresponding expert features for the emotion dataset.

A feature for the emotion dataset is a single word. We choose to not further break down the words into tokens, as it is unclear what the projection of a single token onto the valence-arousal plane would mean. A group is a collection of words in the input that need not appear consecutively.

**Expert Features.** An expert feature is a word that is extremely close to an axis point on the valence arousal plane - see Table 3 or Table 6 for examples of such expert features.

### D.5 Chest X-Ray Dataset

See Figure 11.

### D.6 Laparoscopic Cholecystectomy Surgery Dataset

See Figure 12.

## Appendix E. Adding a New Setting.

Here, we provide a step-by-step walkthrough for adding a new setting to the FIX benchmark, so that the process may be more accessible to future researchers.

1. Determine if the new setting has explicit or implicit expert alignment.

2. If the setting has explicit expert alignment, i.e. there are explicit annotations for expert features available, one can use the explicit's case's ExpertAlign function, as shown in Equation 3.

3. Otherwise, if the setting has implicit expert alignment, one must define a custom expert alignment scoring function for that setting.
   *Note:* We suggest consulting with experts of that domain so that the criteria incorporated in the formulation of the scoring function aligns well with expert judgment.

4. Once the expert alignment scoring function is defined, we can plug this into the FIX framework, as defined in Equations 1 and 2, to obtain the FIXSCORE for the setting.

5. Depending on the data modality of the setting, one can run relevant baseline methods, including those we provide in Section 5.

## Appendix F. Compute Resources

All experiments were conducted on two server machines, each with 8 NVIDIA A100 GPUs and 8 NVIDIA A6000 GPUs, respectively.

## Appendix G. Safeguards

The datasets and models that we use in this work are not high risk and are previously open-source and publicly available. In particular, for our medical settings which would pose the most potential safety concern, the datasets we sourced our FIX datasets from are already open-source and consists of de-anonymized images.

## Appendix H. Datasheets

We follow the documentation framework provided by Gebru et al. (2021) to create datasheets for the FIX datasets. We address each section per dataset.

### H.1 Motivation

**For what purpose was the dataset created?**
- **Mass Maps**: The original dataset, CosmoGridV1 (Kacprzak et al., 2023), was created to help predict the initial states of the universe in cosmology.
- **Supernova**: The original dataset PLAsTiCC for Kaggle competition (Allam Jr et al., 2018), was created to classify astronomical sources that vary with time into different classes.
- **Multilingual Politeness**: The Multilingual Politeness dataset (Havaldar et al., 2023a) was created to holistically explore how politeness varies across different languages.
- **Emotion**: The original dataset, GoEmotions (Demszky et al., 2020), was created to help understand emotion expressed in language.
- **Chest X-Ray**: The NIH-Google dataset (Majkowska et al., 2020), which is a relabeling of the NIH ChestX-ray14 dataset (Wang et al., 2017), was created to help identify the presence of common pathologies.
- **Laparoscopic Cholecystectomy Surgery**: The original datasets from M2CAI16 workflow challenge (Stauder et al., 2016) and Cholec80 (Twinanda et al., 2016) were created to help identify the safe and unsafe areas of surgery.

**Who created the dataset (e.g., which team, research group) and on behalf of which entity (e.g., company, institution, organization)?**

- **Mass Maps**: The original dataset CosmoGridV1 (Kacprzak et al., 2023) was created by Janis Fluri, Tomasz Kacprzak, Aurel Schneider, Alexandre Refregier, and Joachim Stadel at the ETH Zurich and the University of Zurich. The simulations were run at the Swiss Supercomputing Center (CSCS) as part of the project "Measuring Dark Energy with Deep Learning", hosted at ETH Zurich by the IT Services Group of the Department of Physics. We adapt the dataset and add a validation split.
- **Supernova**: The original dataset PLAsTiCC was created by Team et al. (2018). We adapt the dataset, add a validation split, and balance the sets for each class.
- **Multilingual Politeness**: The Multilingual Politeness dataset (Havaldar et al., 2023a) was created by Shreya Havaldar, Matthew Pressimone, Eric Wong, and Lyle Ungar at the University of Pennsylvania.
- **Emotion**: The original GoEmotions (Demszky et al., 2020) dataset was created by Dorottya Demszky, Dana Movshovitz-Attias, Jeongwoo Ko, Alan Cowen, Gaurav Nemade, and Sujith Ravi at Stanford University, Google Research and Amazon Alexa.
- **Chest X-Ray**: The NIH-Google dataset (Majkowska et al., 2020) was created by Anna Majkowska, Sid Mittal, David F Steiner, Joshua J Reicher, Scott Mayer McKinney, Gavin E Duggan, Krish Eswaran, Po-Hsuan Cameron Chen, Yun Liu, Sreenivasa Raju Kalidindi, et al., at Google Health, Stanford Healthcare and Palo Alto Veterans Affairs, Apollo Radiology International, and California Advanced Imaging.
- **Laparoscopic Cholecystectomy Surgery**: The M2CA116 workflow challenge dataset (Stauder et al., 2016) was created by Ralf Stauder, Daniel Ostler, Michael Kranzfelder, Sebastian Koller, Hubertus Feußner, and Nassir Navab at Technische Universität München in Germany and Johns Hopkins University. The Cholec80 dataset (Twinanda et al., 2016) was created by Andru P Twinanda, Sherif Shehata, Didier Mutter, Jacques Marescaux, Michel De Mathelin, and Nicolas Padoy, at ICube, University of Strasbourg, CNRS, IHU, University Hospital of Strasbourg, IRCAD and IHU Strasbourg, France.

**Who funded the creation of the dataset?**

- Please refer to each setting's respective papers for funding details.

### H.2 Composition

- The answers are described in our paper. Please refer to Section 4 and Appendix A for more details.

### H.3 Collection Process

- We defer the collection process to the relevant works that created them. Please refer to Section 4 and Appendix A for more details.

### H.4 Preprocessing/cleaning/labeling

- The answers are described in our paper. Please refer to Section 4 and Appendix A for more details.

### H.5 Uses

- The answers are described in our paper. Please refer to Section 4 and Appendix A for more details.

### H.6 Distribution

**Will the dataset be distributed to third parties outside of the entity (e.g., company, institution, organization) on behalf of which the dataset was created?**
- No. Our datasets will be managed and maintained by our research group.

**How will the dataset will be distributed (e.g., tarball on website, API, GitHub)?**
- The FIX datasets are released to the public and hosted on Huggingface (please refer to links in Appendix A).

**When will the dataset be distributed?**
- The datasets have been released now, in 2024.

**Will the dataset be distributed under a copyright or other intellectual property (IP) license, and/or under applicable terms of use (ToU)?**
- **Mass Maps**: The Mass Maps dataset is distributed under CC BY 4.0, following the original dataset CosmoGridV1 (Kacprzak et al., 2023).
- **Supernova**: The Supernova dataset is distributed under the MIT license.
- **Multilingual Politeness**: The Multilingual Politeness dataset is distributed under the CC-BY-NC license.
- **Emotion**: The Emotion dataset is distributed under the Apache 2.0 license.
- **Chest X-Ray**: The Chest X-Ray dataset is distributed under the Apache 2.0 license.
- **Laparoscopic Cholecystectomy Surgery**: The Laparoscopic Cholecystectomy Surgery dataset is distributed under the CC by NC SA 4.0 license.

## Appendix I. Author Statement

We bear all responsibility for any potential violation of rights, etc., and confirmation of data licenses.

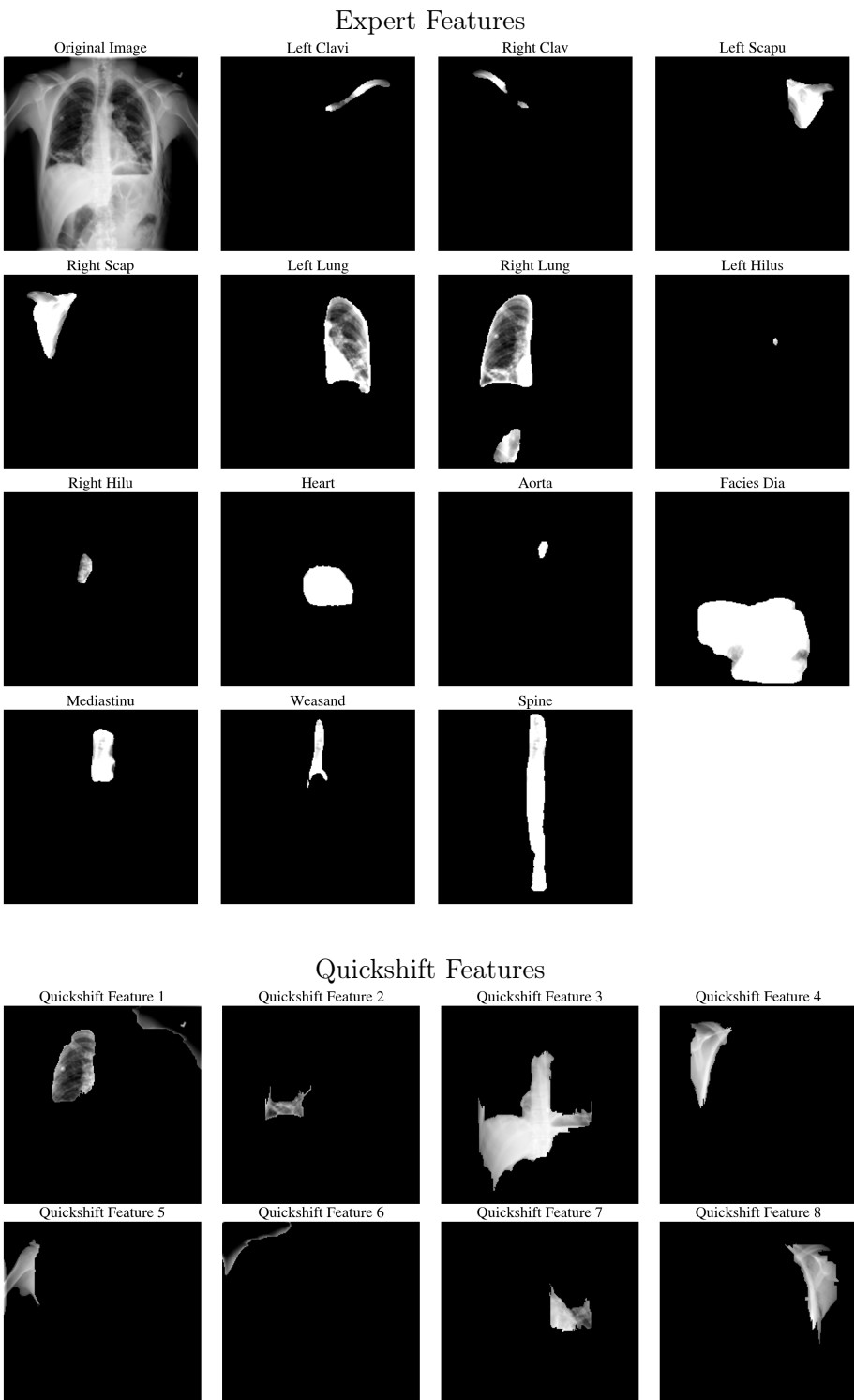

Figure 11: Chest X-ray features from experts (top) and some samples from quickshift (bottom).

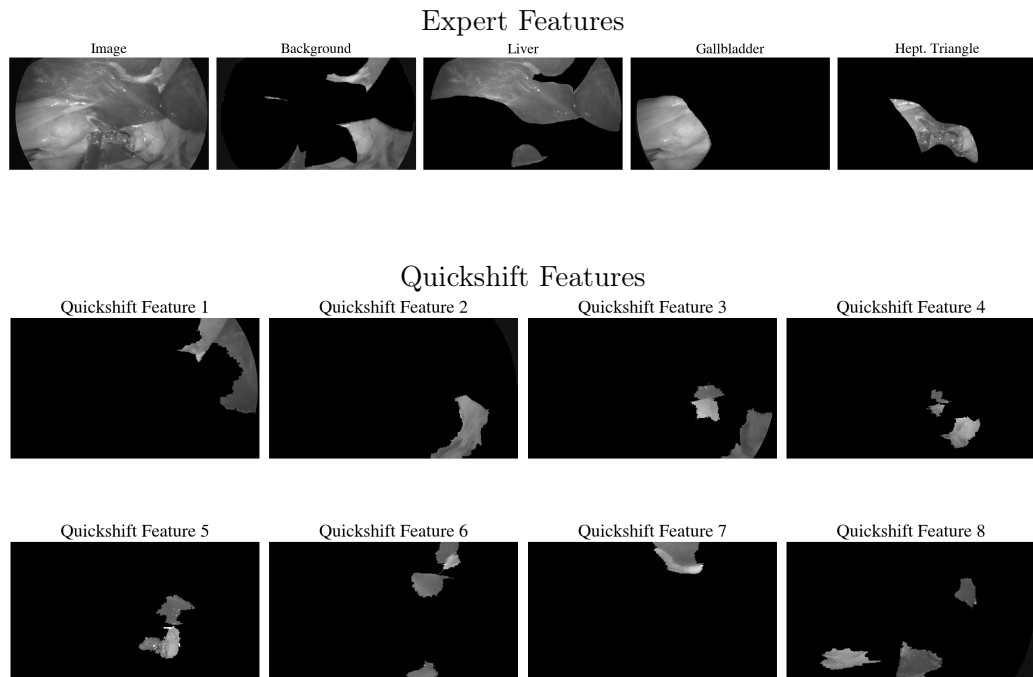

Figure 12: Laparoscopic Cholecystectomy features from experts (top) and some samples from quickshift (bottom).

