# OpenReview forum: "The FIX Benchmark: Extracting Features Interpretable to eXperts"
_DMLR — Accepted by DMLR_

### Review · Reviewer_FBBM · 2025-03-25

**Recommendation:** 3
**Confidence:** 1

**Summary Of Contributions:**

The paper introduces FIX, a new benchmark designed to evaluate how well machine learning models extract features that are understandable by domain experts. While feature-based methods are commonly used to explain model predictions, for high-dimensional data we can't assume the avaliability of interpretable features and it's also challenging to define such important features.


To address this, the authors created FIXScore, a unified measure for expert alignment across different real-world settings in cosmology, psychology, and medicine, covering vision, language, and time series data. Taking a step further, the authors used FIXScore to evaluate popular feature-based explanation methods and found that they don't align well with expert-specified knowledge. This finding highlights the need for new methods that can better identify features that are interpretable to experts.

**Strengths:**

The strengths are listed in the previous section.

**Audience:**

No

**Broader Impact Concerns:**

N.A.

**Claims And Evidence:**

Yes, all the claims seem to be well supported

**Datasets And Benchmarks:**

Yes, the dataset is publically avaliable in HuggingFace. The paper provides sufficient details to support reproducibility.

**Extended Submissions:**

N/A

**Limitations:**

Listed in the "Strengths And Weaknesses" section.

**Requested Changes:**

N.A

**Strengths And Weaknesses:**

Strength:
* The authors clearly define FIX and FIXScore, detailing how it addresses the identified gap by providing a unified measure for expert alignment across different domains.  They also explain how FIXScore is designed to work with both implicit and explicit expert alignment.
* The authors provide detailed descriptions of the datasets used in the benchmark, including the problem setup, the expert features, and the expert alignment function used for each dataset.  This level of detail supports the claim that the benchmark is well-defined and comprehensive.
* The authors present a thorough evaluation of baseline methods using FIXScore.  The results are clearly presented in Table 2, and the authors discuss the implications of these results, supporting their claim that current methods do not align well with expert-specified knowledge.


Weakness
* While the benchmark covers diverse domains, the authors also mention that the metrics for the included domains may not be appropriate for all settings. Also, though they claimed the FIX score is designed "by experts and for experts", it seems like designing such expert features is still time-consuming. To sum up, more discussion about generalizing such methods is really appreciated
* The technical contribution seems to be limited. In my opinion, only the metric of feature alignment (Sec 3.1) is very novel, while other sections remains some descriptions of these existing or well established metrics. Please correct me if I'm wrong.

In summary, the authors provide substantial evidence to back their claims regarding the FIX benchmark, its design, and its potential to evaluate and improve the interpretability of machine learning features. However, since I'm not an expert in the area of explainable AI, I'm not very confident about my evaluation.

---

### Review · Reviewer_rdWK · 2025-04-03

**Recommendation:** 3
**Confidence:** 2

**Summary Of Contributions:**

The paper introduces the FIX benchmark and FIXScore, a unified metric to evaluate how well automatically extracted feature groups align with domain expert knowledge across diverse real-world datasets in cosmology, psychology, and medicine.

**Strengths:**

See Strengths above.

**Audience:**

Yes

**Claims And Evidence:**

Yes

**Datasets And Benchmarks:**

Yes

**Extended Submissions:**

NA

**Requested Changes:**

See Weaknesses above.

**Strengths And Weaknesses:**

Strengths:
1. The paper systematically organizes its contributions, datasets, and experiments, making the flow easy to follow.
2. Figures and tables (e.g., Table 1, Figure 3) effectively illustrate expert features and baseline results.
3. The benchmark addresses real-world needs (e.g., medical image explanations) with input from domain experts.

Weaknesses:
1. In Equation 1, the notation G[i] (groups covering feature i) is not explicitly defined in the text, leading to potential confusion.
2. The implicit vs. explicit ExpertAlign definitions (Equations 4–7 vs. Equation 3) lack a unified justification for their distinct formulations across domains.
3. The Mass Maps purity metric (Equation 10) assumes "void" and "cluster" are mutually exclusive, but real-world maps may contain overlapping structures.
4. Domain-agnostic baselines (e.g., clustering) are simplistic and do not include state-of-the-art methods like diffusion-based segmentation or LLM-driven token grouping.

---

### Review · Reviewer_4FmK · 2025-04-04

**Recommendation:** 4
**Confidence:** 1

**Summary Of Contributions:**

The FIX Benchmark introduces a framework for evaluating how well machine learning features align with expert understanding across diverse domains. The authors developed "FIXScore," a unified metric that measures feature interpretability by evaluating how low-level features (pixels, tokens, time points) are grouped into expert-aligned higher-level concepts. They curated six datasets spanning cosmology, psychology, and medicine across vision, language, and time-series modalities, implementing both implicit alignment (using expert-defined scoring functions) and explicit alignment (measuring overlap with expert annotations) approaches. For each domain, they collaborated with experts to develop specific interpretability criteria: void/cluster purity for cosmological mass maps, linear consistency for supernova time-series, lexical category alignment for politeness assessment, emotional circumplex distance for emotion classification, and anatomical structure overlap for medical imagery. Their evaluation of existing feature extraction methods demonstrated that current approaches perform poorly on expert alignment, highlighting the need for developing new methods specifically designed to extract features that domain experts find interpretable.

**Strengths:**

See above

**Audience:**

Yes

**Claims And Evidence:**

Yes

**Datasets And Benchmarks:**

Yes

**Extended Submissions:**

N/A

**Requested Changes:**

None

**Strengths And Weaknesses:**

I think the paper is ready for publication and I didn't find any flaws.

The paper is especially strong as the authors collect expert data and collaborate to craft the correct metric for the interpretability work on various domains.

Perhaps the authors could provide more documentations around how other researchers would create a new interpretability benchmark for other domains and a step by step guide in creating so.

It could also be beneficial to include a few example features that are extracted by existing methods and how those deviate from expert features.